# Development of a Human-Display Interface with Vibrotactile Feedback for Real-World Assistive Applications

**DOI:** 10.3390/s21020592

**Published:** 2021-01-15

**Authors:** Kiduk Kim, Ji-Hoon Jeong, Jeong-Hyun Cho, Sunghyun Kim, Jeonggoo Kang, Jeha Ryu, Seong-Whan Lee

**Affiliations:** 1Department of Brain and Cognitive Engineering, Korea University, Seoul 02841, Korea; dukekim@korea.ac.kr (K.K.); jh_jeong@korea.ac.kr (J.-H.J.); jh_cho@korea.ac.kr (J.-H.C.); 2R&D Center, LG Display, Seoul 07796, Korea; gjg21c@lgdisplay.com; 3Innosimulation, Seoul 03925, Korea; shkim246@innosim.com; 4School of Integrated Technology, Gwangju Institute of Science and Technology, Gwangju 61005, Korea; ryu@gist.ac.kr; 5Department of Artificial Intelligence, Korea University, Seoul 02841, Korea

**Keywords:** haptics, assistive technology, vibrotactile feedback, active touch, real-world environment

## Abstract

It is important to operate devices with control panels and touch screens assisted by haptic feedback in mobile environments such as driving automobiles and electric power wheelchairs. A lot of consideration is needed to give accurate haptic feedback, especially, presenting clear touch feedback to the elderly and people with reduced sensation is a very critical issue from healthcare and safety perspectives. In this study, we aimed to identify the perceptual characteristics for the frequency and direction of haptic vibration on the touch screen with vehicle-driving vibration and to propose an efficient haptic system based on these characteristics. As a result, we demonstrated that the detection threshold shift decreased at frequencies above 210 Hz due to the contact pressure during active touch, but the detection threshold shift increased at below 210 Hz. We found that the detection thresholds were 0.30–0.45 gpeak with similar sensitivity in the 80–270 Hz range. The haptic system implemented by reflecting the experimental results achieved characteristics suitable for use scenarios in automobiles. Ultimately, it could provide practical guidelines for the development of touch screens to give accurate touch feedback in the real-world environment.

## 1. Introduction

In the real world, we recognize objects or operating status by combining the information obtained through the five senses. As an application combined with the demand for healthcare, touch screens with haptic feedback assist visually impaired people, sensory impaired people, and elders to access information such as steering and control commands for mobile devices through the sense of touch. Moreover, proper rendering in a mobile environment achieves accurate tactile feedback on touch displays, to assist users to manipulate the devices such as wheelchairs and vehicles precisely and safely. The development of haptic technology, which was traditionally an important sense, was relatively unremarkable. However, since touch screens have been used as an input interface for mobile phones, the demand for haptic feedback has increased, and the surface haptics field in haptics has begun to emerge [1,2].

Surface haptics can be categorized into two types depending on implementation methods [3]. The first is the vibrotactile method that delivers direct vibration below 500 Hz between the finger and the touch screen [4]. The second is the method to provide tactile feedback through a friction modulation between them [5,6]. The research results of surface haptics over the last 10 years indicate that more active research is being conducted on friction modulation compared to that on vibrotactile. However, the commercially successful haptic technology is almost the only vibrotactile technology using an Eccentric Rotatory Mass (ERM) motor applied to a mobile phone. Of course, Apple’s products, one of the healthcare applications, with a linear resonant actuator (2016) and the Tanvas’ monitor with electrostatic tactile (2019) were also released, but the market response was not enthusiastic.

As shown in Table 1 [7,8], although an ERM motor is not the actuator with the best characteristics, it has become a major actuator for mobile phones. This is because when using a touch interface on a mobile phone, the need for haptic feedback is reduced because the user is almost always looking at the screen. Instead, the demand for an alarm notifying the reception of a call or a text message is large. This can be said to be a choice made by users in consideration of the use scenario and efficiency of the application to be used, as well as the performance of haptic feedback.

However, it is challenging to implement a haptic usage scenario in an automotive touch screen with ERM motors. In a vehicle, the touch screen must be operated while minimizing distraction. Therefore, it is required to have immediate response and haptic feedback capable of recognizing a boundary or location of buttons and confirming the state of pressing a button. In particular, there have been previous studies on the detection threshold of haptic stimulation for static pressure, but few studies focused on the detection threshold when a certain pressure was exceeded in the active touch operation. In addition, when the touch and the haptic interaction are combined with vibration generated during driving, deriving the haptic detection threshold becomes important for developing an efficient haptic embedded touch screen for automobiles.

In this study, the detection thresholds of haptic stimulation in the normal and tangential directions given from the automotive touch screen are derived for active touch motions in which a specific pressure is applied to the touch screen. The experiments are conducted in an environment with vibration noise generated while driving a vehicle. Based on the derived detection threshold, prototypes of actual haptic embedded touch screens are established. The implementation of haptic systems is not a new attempt because it has already been proposed by Banter et al. [9] as well as applied to mobile phones. The manufacturing of haptic systems is conducted for the purpose of comparing and verifying the advantages and disadvantages of creating and operating the haptic prototypes. This study deals with the vibrotactile method considering the necessary characteristics and efficiency required (Table 1). The purpose of this study is to develop a haptic system that can obtain clear haptic feedback even in real-world environments, such as driving a vehicle for people with weak senses. Although these experiments have been performed using touch screens for vehicles, this study shows the feasibility of developing a healthcare technology applied to essential areas for user safety, such as wheelchair control panels.

## 2. Related Works

Several related studies have reported that haptic feedback improves the response time and performance [10,11,12,13,14,15,16,17,18,19]. In particular, in environments requiring visual attention such as driving, lack of haptic feedback could considerably hinder safety, making such critical feedback [10,20,21].

The perception of haptic stimuli depends on a variety of factors, such as frequency, amplitude, waveform, duration, and contact area. There are many comprehensive studies and review reports on vibrotactile perception [7,16]. It is well known that the detection threshold (i.e., the minimum perceptible intensity) of the haptic stimuli varies with frequency [22,23,24,25]. The detection threshold also depends on the shape of the stimulator and the location on hand where the haptic stimuli are transmitted. In addition, Israr et al. [26] reported that the just noticeable difference (JND) of haptic vibration (i.e., the amount by which it must change for a difference to be detectable) in the presence of interfering vibration is larger than the JND in the absence of interfering vibration. Meanwhile, studies have also been conducted on the duration and latency of stimuli as temporal conditions of haptic feedback in addition to intensity. Zwislocki and Verrillo [27,28,29] conducted psychophysical experiments to report that the detection threshold of haptic vibration decreases by 3 dB when the duration is doubled. David et al. [30] reported the difficulty of users perceiving feedback as being due to their own actions as the delay between motion and feedback increases. However, haptic studies on touch screens have mainly focused on modeling the click vibrations that occur at moments of state transition of a button rather than on basic perceptual characteristics [31]. Since Fukumoto and Sugimura [32] first introduced the “active touch(click)” concept, studies have been conducted on the magnitude, duration, and stimulator for vibration signals to implement a significantly realistic haptic feedback of the press-and-release stroke of a virtual button on a touch screen [33,34,35,36,37,38]. These studies have important implications because pressing a button is preferred over using touch screens in vehicles, where visual distraction can be a problem.

Previous studies based on noise have reported a positive masking effect, where weak haptic stimuli below threshold are perceived more easily by adding a certain intensity of noise vibration, whereas noise vibration can act as a negative masker for suprathreshold haptic vibration [39,40]. However, the noise vibration applied in Collins et al.’s study used digitally generated quasi-white noise, and it was an experiment to see how the noise affected the detection threshold when the intensity of the haptic stimulation was smaller or larger than the threshold. Besides, Ng et al. conducted experiments on the accuracy of operation during actual vehicle driving, for experiments under the driving environment [17]. However, Ng et al. noted that there was a limitation because the haptic stimulation was not sufficiently large compared to the driving vibration, and their experiments were not to find the detection threshold. Since the driving vibration of the actual vehicle was recorded and applied to the experiment, the effect of distraction by driving the vehicle was excluded and the detection threshold was derived in this study.

Finally, the framework of experiments in this study and implementation of the actual haptic system are referred from the studies of Breitschaft et al. and Banter [9,41].

## 3. Materials and Methods

### 3.1. Participants

Ten right-handed participants, seven men and three women with a mean age of 24.3 years, and a standard deviation of 2.0 years, with no sensorimotor impairment, volunteered for the main experiments. Written consent was acquired from all the participants, and the experimental protocol was approved by the Institutional Review Board (IRB) at the Gwangju Institute of Science and Technology (20160616-HR-23-06-04).

### 3.2. Experimental Setup

Figure 1a illustrates the experimental setup for these experiments. The display used for the experiments herein was an automotive full high–definition resolution liquid crystal display (LCD; total dimensions: 261 mm × 110 mm × 9 mm; active area: 255 mm × 103 mm; and weight: 310 g). To drive the display, an interface board transmitted the digital visual interface (DVI) to a personal computer (PC) and a low-voltage differential signaling interface to the LCD. For the timing of application regarding the haptic feedback, information from the touch sensor embedded in the display was used. If the recognized information was transmitted to the PC from those sensors, the PC generated the image information by comparing the touch gesture (location and time) with the GUI location (Figure 1b), drove the display through the interface board, and commanded the haptic stimulation reproduction through the control board. The haptic commands generated by the PC controlled the on/off, offset, frequency, and a repetition period of the signal from the microcontroller unit of the control board and amplified them using an external amplifier to the voltage required for the vibration shaker. Brüel & Kjær model 4809 was used as a shaker, and the amplifier was Brüel & Kjær model 2718. A shaker was fixed to the center of the display’s back to generate the out-of-plane (the normal direction) vibration of the display. The other shaker was placed horizontally beside the short side of the display and fixed to the display device to generate the in-plane (the tangential direction) vibration (Figure 1b).

### 3.3. Preliminary Experiment

Most studies on driving vibration of automobiles have been conducted in relation to the safety and health of drivers [42,43]. With respect to the magnitude of driving vibration, Directive 2002/44/EC standardizes the daily exposure limit value to 1.15 m/s${}^{2}$ because it is known that probable health risk is high when whole-body vibration is greater than 0.9 m/s${}^{2}$. However, these results do not provide us with any insight because the driving vibration required in this study is the vibration on the display, whereas the previous studies on driving vibration concentrated mainly on the vibration felt by the driver’s body.

Figure 2 shows the vibration amplitude in terms of acceleration over time and power spectral density at each frequency for each direction at 80 km/h by sampling for 1/1000 s. We measured the driving vibrations on the display in two vehicles (KIA Sedona and Tucson Ix) with diesel engines running at a constant speed on asphalt roads, using E2BOX’s EBIMU24GV3 acceleration measuring instrument and Crossbow’s VG700CB-200 aircraft acceleration measuring instrument at the Korea Automotive Technology Institute and the Gwangju Institute of Science and Technology, respectively. The measuring setup in Figure 3 shows acceleration measuring instruments used to acquire driving vibration. The driving vibration amplitude was measured five times for 8 s to obtain the average.

The analysis showed that the average driving vibration was around 0.1 grms (1.0 m/s${}^{2}$) in the x- and z-directions and 0.08 grms (0.8 m/s${}^{2}$) in the y-direction. The up-down and front-rear driving vibrations were calculated to be approximately 0.09 grms (0.9 m/s${}^{2}$), smaller than those in the x- or z-direction. Therefore, the z-direction driving vibration, which had the largest vibration magnitude, was used for the experiments because the touch screens would be mounted in the car at an angle that was between the lines perpendicular and parallel to the ground. Furthermore, only the z-direction of vibration was equally applied as noise in the out-of-plane and in-plane vibration experiments to assess how the detection threshold differed between the vibration directions. This simplification was intended to understand the influence of the same driving vibration condition as noise.

### 3.4. Experimental Protocols

Figure 4 illustrates a schematic of the experimental design paradigm. Since this study is the perceptual experiment of haptic feedback for active touch, the contact pressure was maintained using an external pressure sensor and a visual indication before acquiring the detection threshold, and the touch location was determined using an embedded touch sensor. Three experiments were conducted to obtain the detection threshold. Experiments 1 and 2 were conducted to compare the detection thresholds for the normal direction haptic stimulation with and without driving vibration, respectively. Experiment 3 was conducted for haptic stimulation in the tangential direction in an environment with driving vibration. Hence, Experiments 2 and 3 were designed to compare detection thresholds for vibration directions.

The target and reference signals were randomly reproduced for buttons A and B, shown in Figure 1b, to determine the detection threshold for haptic feedback on the touch screen using a two-alternative forced-choice paradigm, and a participant was asked which button was stronger than the other. The one-up two-down adaptive staircase method was used to determine the vibration intensity to be reproduced for the experiment [44,45].

If the participant was accurate in one trial, then the amplitude of the next target remained the same. Subsequently, when the participant accurately chose a target stimulation twice in succession, the amplitude of the next target stimulation was decreased by a predetermined step size. Otherwise, the amplitude of the next target stimulation was increased by the same step size. The initial step size was 1 dB, but after the first three response reversals(the change in stimulation amplitude from decreasing to increasing or vice versa), the step size decreased to 0.25 dB. The session ended after 12 reversals and the average amplitude of the last 7–12 reversals was taken as the detection threshold. Prior to each experiment, a briefing of the task and a short training session were conducted for the participants. Each session lasted approximately 15 min and ended after 30–50 attempts for a total of 40–60 min for each experiment. It was compulsory for the participants to take a break between sessions. During the experiments, white noise was played through headphones or earbuds to block the effects of external sound.

Experiment 1 involved the detection threshold of button-clicking feedback on the touch screen to which the normal direction of haptic stimulation was reproduced. A vibration acceleration of 1 gpeak (9.8 m/s${}^{2}$) for haptic feedback was used as the first target stimulus, because this acceleration could be sufficiently perceived stronger than the reference stimulus based on previous research and preliminary experiment. The reference button was kept in the non-vibration state and the haptic stimulation was applied only to the target button according to the procedure. Experiments 2 and 3 involved the detection threshold in the case of vehicle driving vibration. Experiments 2 and 3 followed the procedure in Experiment 1; however, it applied a driving vibration to the reference button and a driving vibration plus a 1 gpeak sinusoidal vibration as the starting point for the target button.

### 3.5. Detection Threshold Estimation

The detection thresholds of the haptic stimuli from this study are expected to be much larger than the results derived from the literature such as Morioka et al. [22].

First, these are perception experiments using a short pulse instead of continuous haptic stimuli to provide haptic feedback for the active touch, so the value is expected to be larger by approximately 12–17 dB than the detection threshold of the continuous haptic stimuli according to Verrillo’s temporal summation (1) [28].
(1)$$\begin{array}{c} \begin{array}{cc} Detection & \;threshold\;shift\;\left(dB\right)\;=\;13+10log_{10}\frac{10}{t}\\ \; & t=\frac{10^{3}}{f}=pulse\;width\;of\;haptic\;stimulation\;\left(\mathrm{ms}\right)\\ \; & f=frequency\;of\;haptic\;stimulation\;(\mathrm{Hz}). \end{array} \end{array}$$

Second, it is expected that the change in contact pressure and contact area due to the active touch action pressing a button will affect the detection threshold. In the literature, including S. Papetti et al., an increase in contact pressure reported a decrease in the detection threshold, but Morioka et al. reported the opposite result [7,46]. Therefore, the detection threshold change by the contact pressure range in this experiment is expected to be approximately ±4 dB, and the specific effect is to be identified through the experiment.

Third, the detection threshold is expected to increase because of the influence of driving vibration. Collins et al. [39] reported that adding a certain intensity of noise to subthreshold haptic vibrations made the user sensitive to haptic stimulation, whereas adding more than a certain intensity of noise made the user insensitive to haptic stimulation. Since the magnitude of the driving vibration analyzed in Section 3.3 is approximately 0.1 grms, which is more than the haptic detection threshold known in previous studies, the detection threshold of this study can be expected to be larger.

Finally, it has been concluded that there was no significant difference in the direction of vibration in Pacinian corpuscles in previous literature such as Basdogan et al. and Brisben et al. [3,47]. However, Hwang et al. [48] reported that the normal direction is statistically more sensitive than the tangential direction. In making actual automotive haptic embedded touch screens, the vibration direction of the haptic stimuli causes a significant difference from the mechanical point of view. Therefore, it is a very interesting experiment to derive the detection threshold in the vibration direction in an environment with driving vibration on an automotive touch screen.

## 4. Results and Discussion

### 4.1. Detection Threshold

We analyzed statistical differences for the mean of the detection thresholds between frequencies using the SPSS statistics tool. For statistical analysis of the results of experiments, a repeated measures ANOVA was used with Mauchly’s test of sphericity of the data and the factor as the frequency. The validity was set to 0.05, and the null hypothesis of a significant effect was rejected if the resulting *p*-value was less than 0.05.

The detection thresholds of haptic stimuli with the normal direction expressed as acceleration on the automotive touch screen were 0.08–0.11 gpeak (approximately 0.8–1.1 m/s${}^{2}$, and the acceleration values in this paper are the peak values unless otherwise noted) and were almost constant regardless of the frequency, as shown in Figure 5a. There was no statistical difference between the frequencies (Huynh-Feldt: F(2.418, 21.762) = 1.127, *p* = 0.351; Mauchly’s test of sphericity: *p* = 0.012). Meanwhile, the statistical analysis of the detection threshold represented by displacement showed that small displacement was detected sensitively as the frequency increased, and the difference between the frequencies was clearly apparent as *p* < 0.05 (Greenhouse-Geisser, F(1.539, 13.854) = 51.837, *p* = 0.00; Mauchly’s test of sphericity: *p* = 0.000). The detection threshold result represented by displacement was similar to those in many previous studies, including those by Morioka and Griffin. Also, the detection threshold decreased gradually with increasing frequency, the lowest being at 270 Hz.

The detection threshold for haptic stimuli with the normal direction on the touch screen in the presence of driving vibration showed an almost constant acceleration value of 0.30–0.39 gpeak at each frequency as shown in Figure 5b. The detection threshold was larger, but the tendency according to frequency was similar to the results without driving vibration. There was no statistically significant difference between the detection threshold values according to frequency (F(3, 27) = 1.546, *p* = 0.225, Mauchly’s test of sphericity: *p* = 0.071). Statistical analysis of the detection threshold in terms of displacement also showed that the detection threshold decreased as the frequency increased, similar to the experimental result without driving vibration (Greenhouse-Geisser, F(1.026, 9.233) = 15.609, *p* = 0.003; Mauchly’s test of sphericity: *p* = 0.000).

In order to compare the vibration directions of the haptic stimulation, the experiment on the tangential direction of vibration was conducted differently from the previous two experiments. As shown in Figure 5c, the detection threshold represented by acceleration was approximately constant as 0.34–0.45 gpeak (3.3–4.4 m/s${}^{2}$) at each frequency, and there was no significant difference by frequency (Greenhouse-Geisser, F(1.073, 9.660) = 1.115, *p* = 0.322; Mauchly’s test of sphericity: *p* = 0.000). The statistical analysis of the detection threshold in terms of displacement was found to be distinctly perceived as different with frequency (Greenhouse-Geisser, F(1.698, 15.278) = 157.544, *p* = 0.000; Mauchly’s test of sphericity: *p* = 0.004). Therefore, we could summarize the experimental results compared to the records from related studies (Figure 5d).

Moreover, in order to test whether there was a difference in perceptual characteristics according to gender, the detection thresholds for each of the normal and tangential directions obtained in the experiments were divided into the results of male and female participants. And as a result of the repeated measures ANOVA of the detection threshold with frequency as a factor, there was no significant difference in all conditions (Detection threshold for male in the normal direction: Sphericity Assumed, *p* = 0.472, for female in the normal direction: Greenhouse-Geisser, *p* = 0.364, for male in the tangential direction: Greenhouse-Geisser, *p* = 0.330, for female in the tangential direction: Huynh-Feldt, *p* = 0.720). However, when the participants are divided into two groups to analyze the effect of gender, the power of the tests was decreased, especially because the number of samples in the female group was small.

### 4.2. Data Analysis

The detection thresholds derived in this study were significantly higher than those of previous studies such as Morioka et al. and Papetti et al. as expected in Section 3.4. At 270 Hz, the detection threshold of haptic stimulus in the normal direction was 15.3 dB higher than that of Morioka et al. as shown in Figure 5d [22,23,46]. In addition, in the case of driving vibration, the detection threshold was higher for normal and tangential directions by 26.9 and 28.1 dB, respectively than Morioka’s result.

#### 4.2.1. Temporal Summation

The influence of the perceived detection threshold due to the difference in vibration duration can be explained by the temporal summation effect reported by Verrillo [28,29] following the study by Zwislocki [27]. It was reported that when the duration of the haptic stimulation was doubled up to 60 ms, the detection threshold decreased by 3 dB and the temporal summation effect disappeared over 1000 ms. The detection threshold decreased by around 13 dB when the duration was increased from 10 to 1000 ms. Verrillo’s temporal summation (the Equation (Equation 1) in Section 3.5) is displayed as a solid line in black in Figure 6. In this experiment, since the sinusoidal signal of one pulse was used as a haptic stimulation, the pulse width is the inverse of the frequency. Therefore, the haptic stimulation of one cycle for each frequency can be expected to be perceived as less sensitive by −12.0, −13.8, −15.6 and −17.3 dB at 80, 120, 180 and 270 Hz, respectively, compared with the continuous haptic stimulation.

#### 4.2.2. Contact Area and Pressure

A study conducted by Papetti et al. on active touch reported that the larger the contact force and area, the smaller the detection threshold. In the active touch operation, when the contact pressure increased from approximately 1 to 5 N, the area in which the finger contacted the touch screen increased by about 35%. Hence, the influence of the detection threshold by the area would not be significant. This could account for the detection threshold decrement by 2.7 dB when the contact area increased by 7.5 times in Morioka et al.’s study.

Conflicting research results have been reported regarding contact force. As mentioned earlier, the study of Papetti et al. presented the experimental results that the detection threshold decreased by 6.4 dB when the contact force increased from 1.9 to 8 N for 250 Hz haptic stimulation. However, Morioka et al. published a result of when the contact force is increased from 1 to 5 N, the detection threshold of approximately 4.4 dB is increased at 125 Hz, although it is a passive touch result. The result for the 5 N contact pressure used in this experiment is shown as a solid line in blue in Figure 6.

The two lines-black and blue- intersect at approximately 210 Hz, and the Equation (Equation 2) which is modeled to reflect the effect of contact pressure in this experiment has a detection threshold shift greater than the Equation (Equation 1) at frequencies not greater than 210 Hz. On the other hand, at frequencies greater than 210 Hz, the detection threshold shift becomes smaller than the Equation (Equation 1). If the coefficient p, which is related to the contact pressure, is fitted in the Equation (Equation 2), the value is not greater than 1 at 210 Hz or less, and greater than 1 when it exceeds 210 Hz.
(2)$$\begin{array}{c} \begin{array}{cc} Detection & \;threshold\;shift\;\left(dB\right)\;=\;13+10log_{10}\frac{10^{-2}\cdot f}{p}\\ \; & f=frequency\;of\;haptic\;stimulation\;(\mathrm{Hz})\\ \; & p\leq 1,\;for\;f\leq 210\mathrm{Hz},\;and\;p>1,\;for\;f>210\mathrm{Hz}. \end{array} \end{array}$$

#### 4.2.3. Noise Vibration of Driving a Car

The effect of noise vibration, which makes haptic stimulation more sensitive, is called a positive masker, whereas the effect of noise interfering with the perception of haptic stimulation is called a negative masker by Collins et al. [39]. The positive masker effects can be described as activating synapses by a certain magnitude of noise in a haptic vibration state, which does not exceed the threshold that causes the neuron synapses to fire. However, the driving vibration used in the present experiments was approximately 0.1 grms, which was much greater than the detection threshold for continuous haptic stimulation of approximately 0.02 gpeak derived results in the perception experiment without driving vibration and the literature. Therefore, the driving vibration acted as the negative masker. In order to perceive haptic stimulation on the automotive touch screen under the condition of driving vibration, it required a stimulation intensity of 8.8–12.2 dB higher than the detection threshold when there was no driving vibration (Figure 7a).

#### 4.2.4. Vibration Direction of Haptic Stimulation

As shown in Figure 7b, the comparison of mean values indicated that the normal direction was 0.4–2.3 dB smaller than the tangential direction, but the result of this experiment indicated that there was no statistically significant difference in the direction of vibration at all frequencies tested, (Two-sample unpaired *t*-test results between directions at 80 Hz: t(18) = −0.20, *p* = 0.841; at 120 Hz: t(18) = −1.37, *p* = 0.187; at 180 Hz: t(18) = −0.73, *p* = 0.474; at 270 Hz: t(18) = −0.54, *p* = 0.593). This result was consistent with the literature such as Basdogan et al. and Brisben et al., where there was no significant difference in the direction of vibration.

### 4.3. Implementation of Haptic Feedback System

Since there is no statistically significant difference in the detection threshold for the vibration direction, a haptic system that is simple to implement and consumes less electric power can be considered economically more advantageous.

The difference in power consumption to provide the same intensity of haptic stimulation between the vibration directions is due to the difference in attenuation until the stimulation reaches the touch screen surface. The touch screen used herein was approximately 9 mm thick, far less than either of the lateral dimensions, and clamped at all four corners of the enclosure considering the car mount. The enclosure was designed with sufficient thickness and mass to become mechanical ground. So the actuator’s haptic vibration was minimized from being transmitted to the outside of the display. The haptic stimulation generated by the actuators was attenuated and modulated by the boundary condition between the touch screen and the enclosure, and the characteristics of the touch screen. If the actuators were to be attached directly to the bezel of the cover window, close to a finger, to apply vibration to the touch screen, it might be a more effective way to generate haptic vibration. However, the current trend of display technology development is toward decreasing bezel size, so either the actuator is attached to the touch screen’s rear, or a film-type actuator is mounted inside the touch screen.

To analyze attenuation according to the direction of haptic stimulation, numerical simulation was performed using ANSYS to see how much the stimulus from the back of the touch screen was transmitted to the front. The vibration displacement at the surface of the touch screen is shown using different colors, from red (maximum value) to blue (minimum value), when the force of 1 N at 120 Hz is applied in the normal and tangential directions at the rear center of the touch screen (Figure 8). The simulation results show that the vibration displacement that reached the front of the touch screen is 0.35 μm for the normal direction of haptic stimulus and 0.02 μm for the tangential direction, which is much larger when the haptic stimulation is given in the normal direction. It can be seen that the mechanical dimensions and boundary conditions of the touch screen for an automobile form a structure that transmits the normal direction of the haptic stimulus better than it does in the tangential direction.

In order to study the simplicity of implementing the haptic system by vibration directions, we created prototypes. The detection threshold of haptic stimulation with the normal or tangential direction was derived to be approximately 0.30–0.45 gpeak (2.9–4.4 m/s${}^{2}$) in the presence of driving vibration. Therefore, to give sufficient haptic feedback to users, haptic stimulation of approximately 4.5 gpeak (44.1 m/s${}^{2}$) or more, that is, ten times the detection thresholds, is needed for the haptic embedded touch screen for automobiles. However, in addition to the magnitude of stimulation, the actuator should be determined by taking into account response time, bandwidth, power consumption, mechanical simplicity, and so forth. These characteristics differed according to the use scenario of the haptic feedback on the touch screen in the automobile, and the necessary characteristics and the actuator types were compared in Table 1. Consequently, the VCM and the piezo actuator were considered as suitable actuator candidates. We selected Bestar’s BTD26-12-08H9 LF2 as the VCM and MPlus’s PHA379060 as the piezo actuator in consideration of the various characteristics and the magnitude of haptic vibration required for the actuators.

Figure 9 illustrates the haptic feedback system for automotive touch screens created using VCMs and piezo actuators. Figure 9a shows the haptic feedback prototype with the stimulation in the normal direction, which consumed 0.88 W of electric power to generate a 4.5 gpeak haptic intensity at 120 Hz using one VCM. In Section 4, the detection thresholds were 0.32, 0.30 and 0.33 gpeak at 80, 120 and 180 Hz, respectively, and there was no significant difference statistically; therefore, any frequency could be used for the haptic stimulation. However, this study determined 120 Hz as the stimulation frequency, which was in agreement with the result obtained by S. Kim [37], showing a higher preference for 120 Hz sine waves than other frequencies by participants. Meanwhile, the haptic system with the tangential vibration shown in Figure 9b,c required the vibration transmission structure to engage the touch screen and VCMs. The tangential vibration system required two VCMs to obtain the same intensity as the normal vibration system and consumed 4.96 W of electric power. As a result of numerical analysis, the prototype with haptic stimulation in the normal direction produced using the VCM consumed less electric power than that with the stimulation in the tangential direction. Figure 9d,e show haptic systems made using piezo actuators and Figure 9f shows measuring the magnitude of the haptic stimulation by attaching accelerometers to the touch screen.

Therefore, it can be seen that the mechanical dimensions and boundary conditions of the touch screen for an automobile form a structure that transmits the haptic stimulation in the normal direction better than it does in the tangential direction. In addition, considering the mechanical size of the actuator used in the implementation of feasible haptic systems, the structure for the tangential direction is thicker than that for the normal direction.

## 5. Conclusions and Future Works

We implemented the haptic embedded touch screen system that is well perceived in driving conditions by establishing the detection threshold of haptic stimulation on the touch screen. Since the temporal summation effect of the literature still fits in the automotive touch screen, the intensity of the haptic stimulation can be modified in consideration of the detection threshold shift according to the duration of the haptic stimulation when designing. However, the findings of this study show that the detection threshold shift decreases due to the contact pressure at frequencies above 210 Hz, but the detection threshold shift increases at below 210 Hz. Furthermore, the detection threshold with driving vibration is 8.8–12.2 dB larger than that without driving vibration. Therefore, 3.0–4.5 gpeak is derived as the intensity of haptic stimulation required for the automotive touch screen under driving conditions.

Ultimately, the study concluded that applying the haptic stimulation in the normal direction is more efficient than in the tangential direction when considering the perceptual and mechanical vibration characteristics of the display to implement the haptic system for automobiles. Although there was no statistical difference in the sensitivity between the vibration directions, the average value of the detection threshold in the normal direction was lower than that in the tangential direction. Moreover, the result indicates that the stimulation in the normal direction has less attenuation than the stimulation in the tangential direction so that haptic stimulation is better transmitted from the actuator to the surface of the touch screen and consumes less electric power.

This study has mainly focused on vibrotactile but we will further research and develop more advanced, practical haptic embedded systems by studying friction modulated tactile haptics and new actuators suitable for touch screens, which are of interest to more researchers than vibrotactile, in the future. Moreover, many studies have been exploring brain activity related to psychophysical characteristics of haptics including emotion and preference. We expect that haptic stimulation could be more deeply analyzed and studied in the neurophysiological domain [49,50,51,52] and neuro-applications [53,54]. Also, the demands on haptic systems that can give clear feedback on human-display interaction even in real-world environments will increase from a healthcare perspective. This will help people with tactile impairment operate their devices more safely and accurately. Hence, we will expand the haptic research on touch screens applied not only to automobiles but also to various real-world assistive applications.

## Figures and Tables

**Figure 1 sensors-21-00592-f001:**
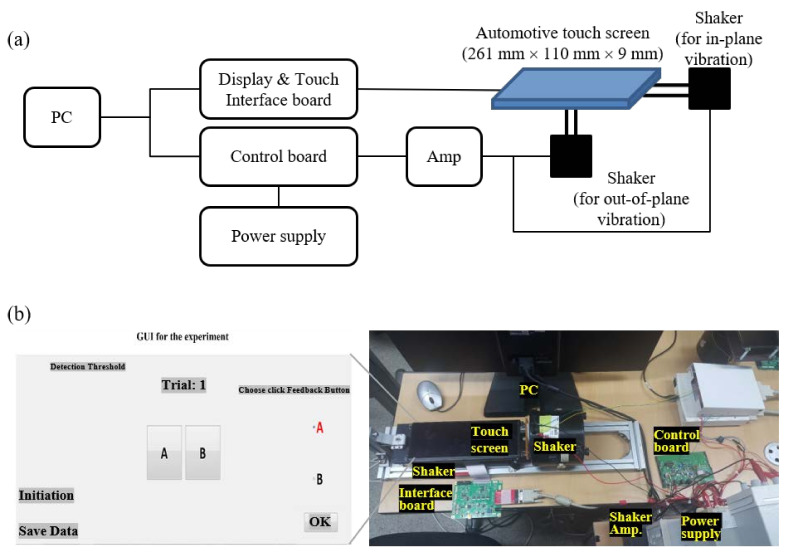
Experimental setup. (**a**) Schematic representation of the experimental setup. (**b**) Image of the experimental setup and GUI.

**Figure 2 sensors-21-00592-f002:**
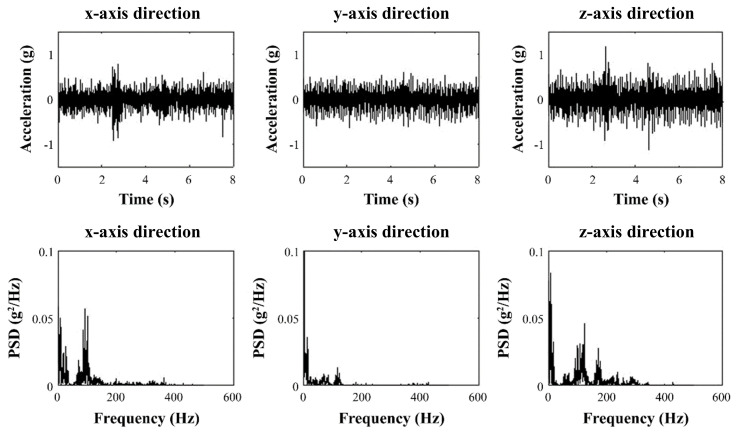
Driving vibration in a time domain and power spectral density of driving vibration.

**Figure 3 sensors-21-00592-f003:**
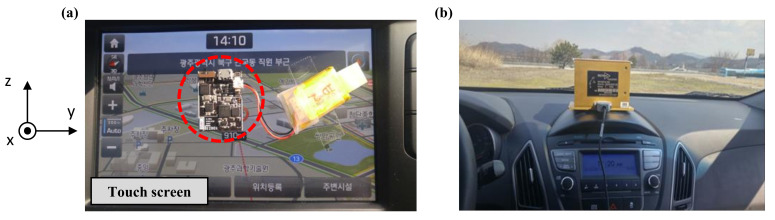
Measuring setup of driving vibration on the touch screen in vehicles. (**a**) E2BOX’s EBIMU24GV3. (**b**) Crossbow’s VG700CB-200.

**Figure 4 sensors-21-00592-f004:**
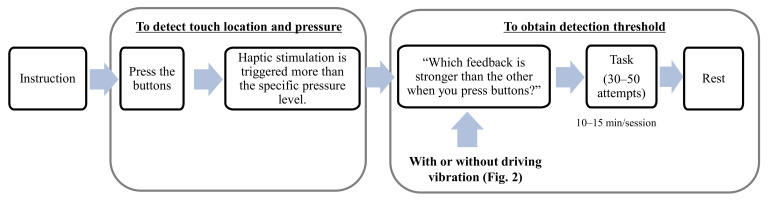
Experimental design paradigm for data acquisition.

**Figure 5 sensors-21-00592-f005:**
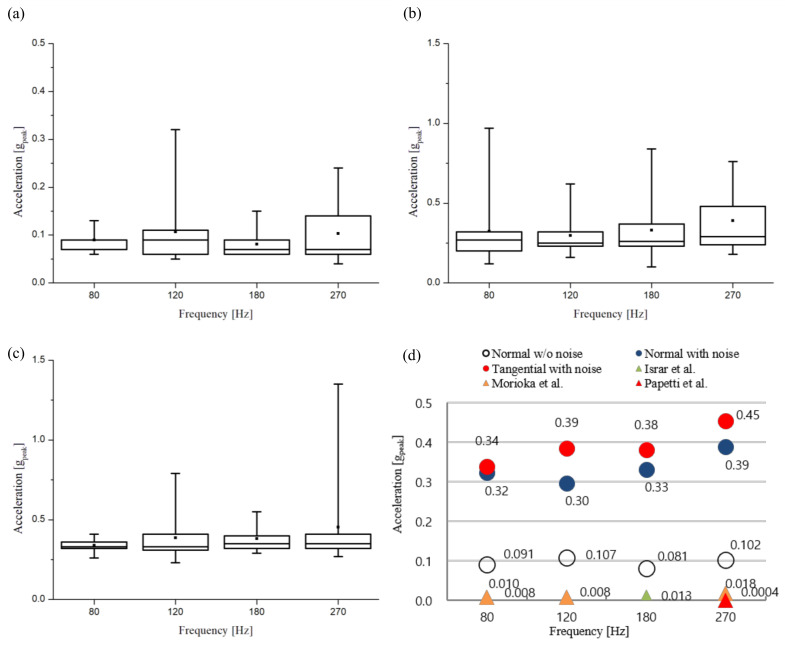
Experimental results of the detection threshold expressed as acceleration. (**a**) For the normal direction without driving vibration. (**b**) For the normal direction with driving vibration. (**c**) For the tangential direction with driving vibration. (**d**) Comparison of experimental results and previously published data.

**Figure 6 sensors-21-00592-f006:**
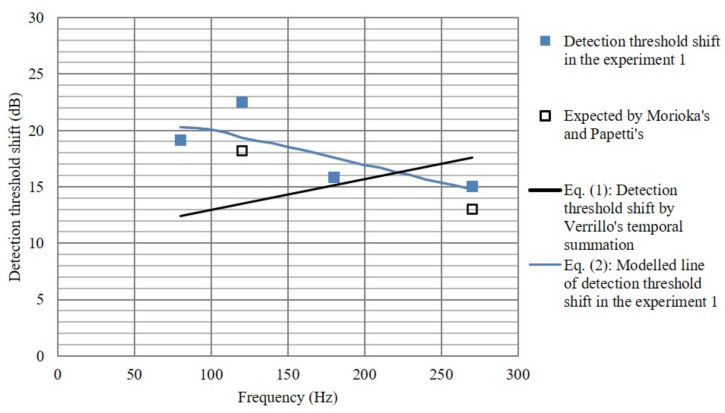
A shift in the detection threshold according to the frequency of the haptic stimulation (Solid line in black: Verrillo’s experimental results; Solid line in blue: Modelled line of detection threshold shift in experiment 1; Closed symbols: Detection threshold shift in experiment 1; Open symbols: Expected shift by contact pressure in the literature).

**Figure 7 sensors-21-00592-f007:**
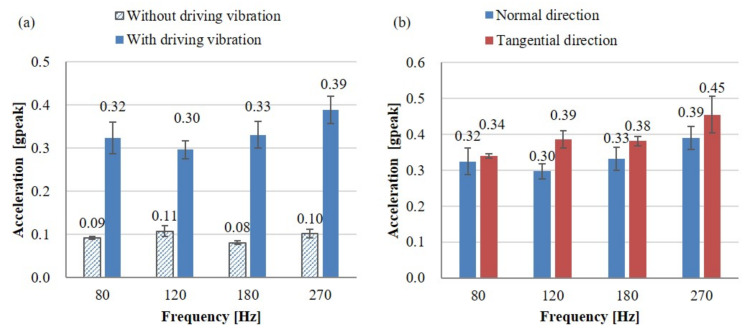
Comparison of the detection thresholds expressed as acceleration. (**a**) Comparison of detection thresholds of haptic stimulation in the normal direction between with and without driving vibration. (**b**) Comparison of detection thresholds of haptic stimulation with driving vibration between the normal and tangential directions.

**Figure 8 sensors-21-00592-f008:**
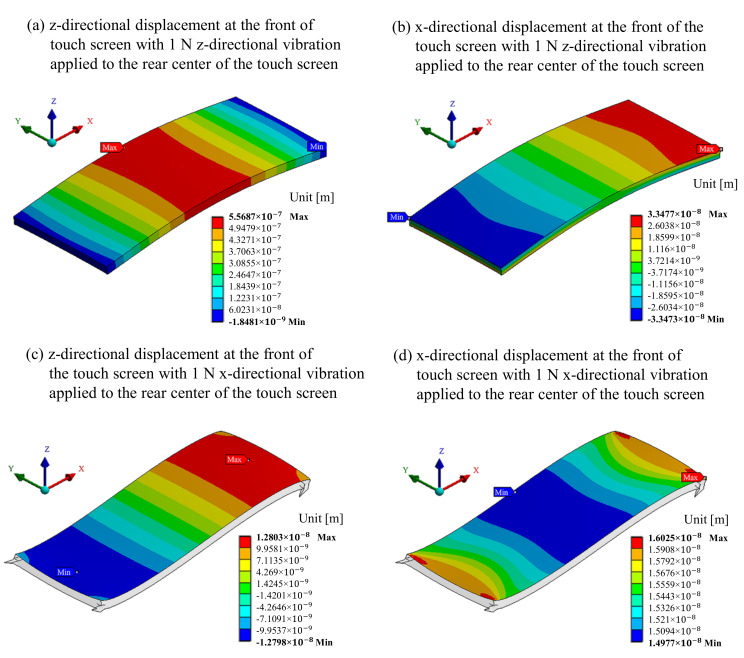
Vibrational displacement on the surface of the touch screen according to the vibration direction of the actuator.

**Figure 9 sensors-21-00592-f009:**
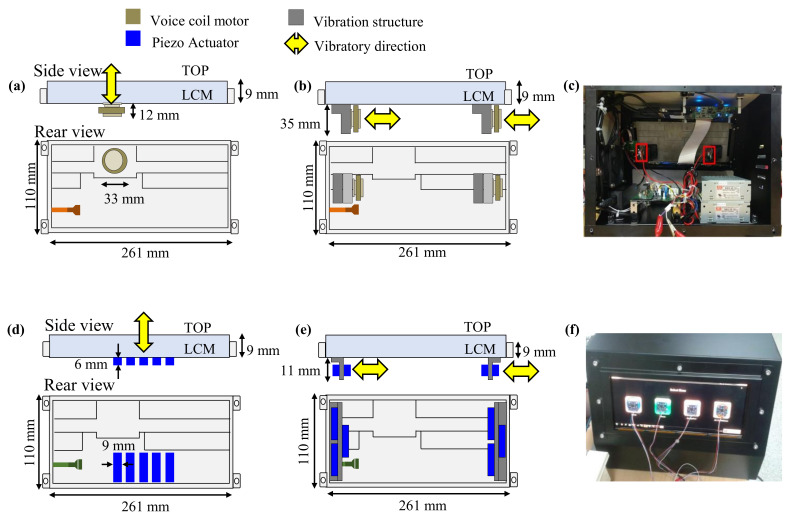
Overview of the haptic systems implemented for automotive touch screens. (**a**) Vibration in the normal direction using a voice coil motor (VCM). (**b**) Vibration in the tangential direction using VCMs. (**c**) Real image of the haptic system vibrated in the tangential direction using VCMs. (**d**) Vibration in the normal direction using piezoelectric actuators. (**e**) Vibration in the tangential direction using piezoelectric actuators. (**f**) Feasibility sample measuring the vibration acceleration.

**Table 1 sensors-21-00592-t001:** Comparison of actuators and needed characteristics for haptic feedback.

	Actuators	Voice Coil Motor	Eccentric Rotating Mass Motor	Linear Resonant Actuator	Piezoelectric Actuator	Electrostatic Tactile Actuator
Characteristics						
**Response** **time**		High	Low	High	High	High
**Bandwidth**		High	Low	Low	High	High
**Displacement** **output**		High	High	Medium	Medium	Low
**Power** **consumption**		Low	Low	Low	High	High
**Mechanical** **simplicity**		Medium	Low	High	Medium	Medium

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
