# Peer review of "Development of a Human-Display Interface with Vibrotactile Feedback for Real-World Assistive Applications"

_sensors, 2021, doi:10.3390/s21020592_

Round 1

Reviewer 1 Report

In the revised manuscript, it seems that the reviewer's points have been well complemented. However, there seems to be a part that needs additional supplementation.

  1. The most essential content in this paper seems to have been confirmed through an experiment on the detection threshold that can recognize haptic feedback when considering the vibration of an actual vehicle while driving. The haptic system using VCM and Piezo actuator presented in Section 4.3 seems to deviate from the main contents of this paper. Haptic systems using VCM and piezo actuators have already been studied for a long time, and experiments or verifications using the manufactured system have not been performed, so it is considered that the contents may be deleted in this paper. In addition, in the case of piezo actuators, it is thought that a resonance phenomenon should be used to generate a large operating force, but explanation of such parts seems to be omitted.
  2. The frequency band in which humans can feel the vibration well is known as 200~300 Hz, but the results in Figure 7 show that the difference according to the frequency is not large. It seems necessary to explain this.

Reviewer 2 Report

The present manuscript shows the design and development of a tablet display with haptic feedback, in particular vibrotactile stimulation. The device has to operate in automotive scenario or, in general, during driving condition (e.g. wheelchair). For such reason, a series of experiment to evaluate the detection threshold of vibrotactile stimulation of ten able bodied participants was developed. Four different frequencies (i.e. 80, 120, 180, 270 Hz) of vibration was tested. Three conditions were compared: vibration delivered in two alternative directions (i.e. normal and tangential) together with mechanical noise versus control condition where only vibration in normal direction was delivered to the participant. The authors found that, in line with literature, the introduction of the noise increases the detection threshold. Additionally, they found that the increase of frequency decreases. The authors also employ a finite elements model analysis to study the mechanical stress-strain behavior on the tablet when vibrotactile stimulation is applied. The theme is of interest, however, the manuscript in some parts resulted to be confused. The manuscript should be partly reorganized and some parts in methods and should be better explained. See the points below.

Major points

Experimental protocols resulted no clear and few information are provided. From figure 2, the first part of the protocol seems to include a touch and location detection stage that is not explained. Additionally, in the main text, it is not explained which it the task of the participant (i.e. identifying the stronger stimulation). How were the participants placed with respect to the setup?

From row 148 to 171, this part is relative to the evaluation of the noise produced by driving, this should be presented prior of the protocols part because it is more similar to a preliminary study for the identification of the noise parameters employed in the experimental protocols.

In “detection threshold estimation” paragraph it is not clear what t is. This is defined as the duration of haptic stimulation, but I guess that is no the duration of the stimulation but the pulse width of the stimulation. This should be better specified.

In Figure 6, I suggest to introduce not only the value of expected detection threshold and the line modelled  on the result of the experiment, but also the values relative to the results obtained in the authors’ work.

In the manuscript, it is not clear how the limit value of 210 Hz was extracted by the authors, an explanation should be included, considering that it is  shown as one of the main findings.

The performed statistical analysis is not clear. The authors employed parametric analysis but they did not claim about the test employed to evaluate the normality of the data distribution. In addition, what do the authors intend with two samples t-test? paired or unpaired test? From the presented data it seems to me that the authors employ the unpaired t-test; but, in some cases, this is not correct. Also when the authors employed the ANOVA, it is not specified which type of ANOVA (e.g. repeated measures, which factors?)? How is the result at line 294 (i.e. “t(56) = -4.65, p = 0.00”) obtained? Please specify.

Minor points

Page 2 row 56 “hatic” should be “haptic”

Page 3, row 89 “below subthreshold” should be “below threshold”

Page 9, row 249,  “A Study” should be “A study”.

Round 2

Reviewer 1 Report

In the revised manuscript, the reviewer's comments have been well complemented and the quality of the paper have been much improved. 

Author Response

Thanks to the valuable comments from the reviewer, the quality of the manuscript has been greatly improved.

Reviewer 2 Report

The manuscript resulted to be improved after the revision. However I have other comments and suggestions:

The ANOVA employed in such work should be a repeated measures one also the sphericity of the data should be evaluated.

Splitting the participants in two groups decreases the power of the tests, the not significant difference between groups could be due to the low number of samples. Please specify it in the text.

Author Response

This manuscript is a resubmission of an earlier submission. The following is a list of the peer review reports and author responses from that submission.

Round 1

Reviewer 1 Report

Authors presents the results of the research taken to identify the perceptual characteristics of the haptic feedback for the active touch on automotive touch screens. They have taken into consideration the influence of vibration of the touch screen and part of the automobile cockpit on the design process of the haptic system, that effects on efficiency of proposed haptic system.

Authors did not presents the differences in the acting on the touch screen between female or male, if there is any difference in feeling? I wonder if this has any influence on the experiment results, cause they selected 7 man and three women for the test.

The review of the literature corresponds to research conducted and all the results are clearly presented. However a few minor mistakes occurred:

In Line 97 – grammar- Fig 1(a) presents?

Figure 1b is unclear, please correct the descriptions (color, font)

Line 114 Fig is not a schematic… Figure presents a schematic…

Correct the quality of figure 5 a,b,c

The presented FEM results of the deflections are without the unit. Please provide the unit for results in countour lines(is it an SI unit [m] or [mm]?

Reviewer 2 Report

The paper shows the characteristics of haptic feedback on active touch with driving conditions. The detection threshold was correlated with duration of haptic stimulation.

The experimental setup was well organized to support the argument.

The authors tested normal and tangential directions and applying haptic stimulation in the normal direction is more efficient. This result is quite different as expected from my view. The authors say there was no statistical difference in the sensitivity between the vibration directions. Often the tangential direction in the movement is more critical to improve the tactile sense.

The stimulation in the normal direction has less attenuation with their haptic platform. I am wondering if the system does not fully support the tactile senses. More experiments with tangential directions with / without driving vibrations seem to be needed. The authors could add experimental conditions or more experiments to explain why senses in the tangential directions are not much supportable.

Reviewer 3 Report

1. The originality of this study should be clearly explained. It is difficult to find differences in this paper compared to previous research works. The proposed haptic embedded touch screen system does not appear to be the first proposed system. 

2. What are the academic advances of this paper related to the field of haptic systems?